# Derandomizing Multi-Distribution Learning

**Kasper Green Larsen**
Department of Computer Science
Aarhus University
larsen@cs.au.dk

**Omar Montasser**[*]
Department of Statistics and Data Science
Yale University
omar.montasser@yale.edu

**Nikita Zhivotovskiy**
Department of Statistics
University of California, Berkeley
zhivotovskiy@berkeley.edu

## Abstract

Multi-distribution or collaborative learning involves learning a single predictor that works well across multiple data distributions, using samples from each during training. Recent research on multi-distribution learning, focusing on binary loss and finite VC dimension classes, has shown near-optimal sample complexity that is achieved with oracle efficient algorithms. That is, these algorithms are computationally efficient given an efficient ERM for the class. Unlike in classical PAC learning, where the optimal sample complexity is achieved with deterministic predictors, current multi-distribution learning algorithms output randomized predictors. This raises the question: can these algorithms be derandomized to produce a deterministic predictor for multiple distributions? Through a reduction to discrepancy minimization, we show that derandomizing multi-distribution learning is computationally hard, even when ERM is computationally efficient. On the positive side, we identify a structural condition enabling an efficient black-box reduction, converting existing randomized multi-distribution predictors into deterministic ones.

## 1 Introduction

We consider the problem of multi-distribution learning where there are $k$ unknown data distributions $\mathcal{P} = \{\mathcal{D}_1, \ldots, \mathcal{D}_k\}$ over $\mathcal{X} \times \{-1, 1\}$, where $\mathcal{X}$ is an input domain and $\{-1, 1\}$ are the possible labels. The goal is to learn a classifier $f : \mathcal{X} \to \{-1, 1\}$ that satisfies

$$\operatorname{er}_{\mathcal{P}}(f) := \max_i \operatorname{er}_{\mathcal{D}_i}(f) \leq \min_{h \in \mathcal{H}} \max_i \operatorname{er}_{\mathcal{D}_i}(h) + \varepsilon, \text{ where } \operatorname{er}_{\mathcal{D}_i}(f) = \Pr_{(x,y) \sim \mathcal{D}_i}[f(x) \neq y]. \quad (1)$$

Here $\mathcal{H} \subseteq \{-1, 1\}^{\mathcal{X}}$ is the benchmark hypothesis class of VC-dimension $d$ that the learner competes against, and $\min_{h \in \mathcal{H}} \max_i \operatorname{er}_{\mathcal{D}_i}(h)$ is the optimal worst-case error that can be achieved with classifiers from $\mathcal{H}$. The framework of multi-distribution learning, introduced by Haghtalab et al. [10], is a natural generalization of agnostic PAC learning [22, 21, 5], and captures several important applications such as min-max fairness [12, 18, 15, 8, 20], and group distributionally robust optimization [16].

In the *realizable* setting, where $\min_{h \in \mathcal{H}} \operatorname{er}_{\mathcal{P}}(h) = 0$, there is a learning algorithm using $\tilde{O}((d+k)/\varepsilon)$ samples to produce such a deterministic classifier $f$, see e.g., the works [4, 7, 13]. Here, and throughout the paper, $\tilde{O}$ hides terms that are $\operatorname{poly} \ln(dk/(\varepsilon\delta))$.

---

[*]This work was primarily done while the author was a FODSI-Simons postdoc at UC Berkeley.

38th Conference on Neural Information Processing Systems (NeurIPS 2024).

In the more challenging *agnostic* setting, where $\text{OPT} := \min_{h \in \mathcal{H}} \text{er}_{\mathcal{P}}(h)$ is greater than 0, recent works show that the sample complexity is $\tilde{O}((d+k)/\varepsilon^2)$ [10, 2, 14, 23]. We refer the reader to Table 1 in [23] for a detailed sample complexity comparison of prior algorithms. Importantly, the guarantee provided by all existing algorithms is slightly different from the objective (1) above. Concretely, all previous algorithms do *not* produce a deterministic classifier $f : \mathcal{X} \to \{-1, 1\}$, but instead output a distribution $F$ over $\mathcal{H}$, such that

$$\max_i \mathbb{E}_{f \sim F}[\text{er}_{\mathcal{D}_i}(f)] \le \min_{h \in \mathcal{H}} \max_i \text{er}_{\mathcal{D}_i}(h) + \varepsilon. \tag{2}$$

Due to the fact that classical PAC bounds, which involve learning from a single distribution, are achieved using deterministic predictors it is somewhat unsatisfactory to always output a randomized predictor in the multi-distribution case. Observe that because, as in (2), we want optimal performance simultaneously for all distributions, even using a randomized algorithm is somewhat problematic. Indeed, assume that in practice we want to sample a single $\hat{f}$ according to $F$ and use it as our predictor. Now, if we seek a guarantee like (1) for $\hat{f}$, then the best we can guarantee from (2) is to use Markov's inequality and a union bound over all $k$ distributions to ensure that

$$\text{er}_{\mathcal{D}_i}(\hat{f}) \le 2k \left( \min_{h \in \mathcal{H}} \max_i \text{er}_{\mathcal{D}_i}(h) + \epsilon \right),$$

with probability at least $1/2$, which is, of course, too conservative. Let us also remark that there are examples of distributions $F$ for which this is basically tight. Consider e.g. an input domain $\mathcal{X} = x_1, \dots, x_k$ and $k$ hypotheses $h_1, \dots, h_k$ such that $h_i(x_i) = -1$ and $h_i(x_j) = 1$ for $j \ne i$. Let $\mathcal{D}_i$ be the distribution that returns $(x_i, 1)$ with probability 1. Then for the uniform distribution $F = k^{-1} \sum_i h_i$ over classifiers, we have $\max_i \mathbb{E}_{f \sim F}[\text{er}_{\mathcal{D}_i}(f)] = 1/k$, but for any single $f$ in the support of $F$, we have $\text{er}_P(f) = \max_i \text{er}_{\mathcal{D}_i}(f) = 1$. The example also shows that for every fixed distribution $\mathcal{D}_i$, if we sample an $f$ from $F$, then with probability $1/k$, its error exceeds the expectation by a factor $k$ for that distribution $\mathcal{D}_i$. There may thus be a large gap between the guarantees of a deterministic and randomized classifier, i.e. the bounds in (1) and (2) are quite different.

The main focus of our work, is on replacing the random classifiers in previous works on agnostic multi-distribution learning by deterministic classifiers and understanding the inherent complexity of doing so. In particular, we are interested in understanding any inherent statistical or computational gaps in multi-distribution learning between deterministic classifiers and randomized classifiers.

**Our contributions**

Our first contribution is a strong negative result towards derandomizing previous classifiers. Recall that the complexity class BPP denotes bounded-error probabilistic polynomial time[2]. That is, problems that have polynomial time randomized algorithms that are correct with probability at least $2/3$ on every input. It is conjectured that $\text{P} = \text{BPP}$ and thus most likely $\text{BPP} \ne \text{NP}$. Recall that a set of $n$ points is shattered if each of the $2^n$ possible labelings of the points can be realized by some $h \in \mathcal{H}$. Our negative result is then the following.

**Theorem 1.** *If* $\text{BPP} \ne \text{NP}$*, then as* $n = \min\{d, k, 1/\varepsilon\}$ *tends to infinity, for* every *hypothesis class* $\mathcal{H}$ *of VC-dimension* $d$ *for which one can find* $n$ *points shattered by* $\mathcal{H}$ *in polynomial time, any multi-distribution learning algorithm for* $\mathcal{H}$ *that on the set of* $k$ *input distributions* $\mathcal{P} = \{\mathcal{D}_1, \dots, \mathcal{D}_k\}$ *with probability at least* $2/3$ *produces a deterministic classifier* $f : \mathcal{X} \to \{-1, 1\}$ *with* $\text{er}_{\mathcal{P}}(f) \le \min_{h \in \mathcal{H}} \text{er}_{\mathcal{P}}(h) + \varepsilon$*, must have either* $n^{\omega(1)}$ *(i.e. super-polynomial) training time, or* $f$ *has* $n^{\omega(1)}$ *evaluation time.*

We remark that this computational hardness result holds even when the class $\mathcal{H}$ admits efficient Empirical Risk Minimization (ERM), and even when the distributions *are known* to the learning algorithm. This highlights that the hardness stems not from the need to sample from the underlying distributions nor from the hardness of ERM, but from the computational problem of deciding which label to assign the points of the input domain.

Note that the assumption in Theorem 1 that one can find a set of $n$ shattered points in polynomial time is not restrictive. Finding such points is trivial for many $\mathcal{H}$, i.e., simply choose $0, e_1, \dots, e_{d-1} \in \mathbb{R}^{d-1} = \mathcal{X}$ for linear classifiers with VC-dimension $d$. More generally, the standard result on the

---

[2]We refer to the monograph [19] as a standard reference discussing computational complexity classes.

class of classifiers induced by positive halfspaces in $\mathbb{R}^d$ shows that this class has VC dimension $d$, and for any set of points such that at most $d$ of its points are contained on a single hyperplane, any subset of size $d$ of this set is shattered. Similar properties are also known for the classes induced by balls in $\mathbb{R}^p$ and positive sets in the plane defined by polynomials of degree at most $p - 1$. See [9] for a detailed exposition of these examples.

While this might have been the end of the story, our NP-hardness proof fortunately highlights a path to circumventing the lower bound. In particular, the proof carefully uses data distributions $\mathcal{D}_1, \ldots, \mathcal{D}_k$ for which $\mathcal{D}_i(y \mid x)$ varies between the distributions. Here $\mathcal{D}_i(y \mid x)$ denotes the conditional distribution of the label $y$ of a sample $(x, y)$ given $x \in \mathcal{X}$. We thus consider the following restricted version of collaborative learning in which $\mathcal{D}_i(y \mid x) = \mathcal{D}_j(y \mid x)$ for all $x, i, j$. That is, the $k$ different distributions may vary arbitrarily over $\mathcal{X}$, but the label $y$ of any $x \in \mathcal{X}$ follows the same distribution for all $\mathcal{D}_i$. As a particular model of *label consistent learning*, one may think of a deterministic labeling setup where it is assumed that there is $f^\star : \mathcal{X} \to \{1, -1\}$ such that across all distributions $y = f^\star(x)$, while no assumption is made that $f^\star$ belongs to $\mathcal{H}$. Remarkably, in terms of sample complexity, in the case of a single distribution, the case of deterministic labeling is almost as hard as the general agnostic case as shown in [3]. Thus, we believe our label-consistent multi-distribution learning setup is quite natural and interesting.

Furthermore, this restriction turns out to be sufficient for derandomizing multi-distribution learning algorithms. In particular, we give a new algorithm, Algorithm 1, that uses a randomized (i.e., an algorithm that outputs a randomized predictor given the training data) multi-distribution learning algorithm (like (2)) as a black-box, and produces from it a deterministic classifier, as in (1).

**Theorem 2.** *For any finite domain $\mathcal{X}$, if the data distributions $\mathcal{D}_1, \ldots, \mathcal{D}_k$ are label-consistent, then given a multi-distribution learning algorithm $\mathcal{A}$ that uses $m(k, d, \mathrm{OPT}, \varepsilon, \delta)$ samples and $t(k, d, \mathrm{OPT}, \varepsilon, \delta)$ training time to produce, with probability $1 - \delta$, a distribution $F$ over classifiers from $\mathcal{H}$ satisfying $\max_i \mathbb{E}_{f \sim F}[\mathrm{er}_{\mathcal{D}_i}(f)] \leq \mathrm{OPT} + \varepsilon$, Algorithm 1 produces with probability $1 - \delta$ a classifier $f : \mathcal{X} \to \{-1, 1\}$ with $\mathrm{er}_{\mathcal{P}}(f) \leq \mathrm{OPT} + \varepsilon$ with the sample complexity*

$$m(k, d, \mathrm{OPT}, \varepsilon/2, \delta/2) + O(k \ln^2(k/(\varepsilon\delta))/\varepsilon^2).$$

*Using the additional ideas in Section 3.1, the training time of Algorithm 1 is*

$$t(k, d, \mathrm{OPT}, \varepsilon/2, \delta/2) + \tilde{O}(k/\varepsilon^2 + \ln(|\mathcal{X}|/\delta)).$$

*If the evaluation time of hypotheses in $\mathcal{H}$ is bounded by $s$, then the evaluation time of the classifier $f$ is bounded by*

$$O(s|F|) + \tilde{O}(\ln(k/\delta) \ln(|\mathcal{X}|/\varepsilon)).$$

Note that several of the previous randomized multi-distribution learning algorithms are indeed computationally efficient as long as ERM is efficient over $\mathcal{H}$. This includes the algorithm in [23] that has a near-optimal sample complexity of $m(k, d, \mathrm{OPT}, \varepsilon, \delta) = \tilde{O}((d + k)/\varepsilon^2)$ with $t(k, d, \mathrm{OPT}, \varepsilon, \delta) = \mathrm{poly}(k, d, \varepsilon^{-1}, \ln(1/\delta)) t_{ERM}$ and $|F| = \mathrm{poly}(k, d, \varepsilon^{-1}, \ln(1/\delta))$, where $t_{ERM}$ denotes the time complexity of ERM over $\mathcal{H}$. Plugging this into Theorem 2 gives a polynomial time deterministic multi-distribution learning algorithm.

We view the restriction to finite domains $\mathcal{X}$ in Theorem 2 as rather mild, as any realistic implementation of a learning algorithm requires an input representation that can be stored on a computer. Moreover, our running time dependency on $|\mathcal{X}|$ is only logarithmic. Even so, in Section 3.2 we give some initially promising directions for extending our algorithm to infinite $\mathcal{X}$.

**Discussion of implications.** Prior work has shown that in agnostic multi-distribution learning, a sample complexity of $\Omega(dk/\epsilon^2)$, which is worse than the optimal $\tilde{O}((d+k)/\epsilon^2)$ sample complexity, is unavoidable with *proper* learning algorithms, which are algorithms restricted to outputting a classifier in the class $\mathcal{H}$ [23, Theorem 18]. In contrast, our negative result in Theorem 1 implies that there is *no* sample-efficient and oracle-efficient multi-distribution learning algorithm that aggregates multiple ERM predictors in polynomial time. For example, our result rules out the simple majority-vote aggregation approach (which is feasible in the realizable setting when $\mathrm{OPT} = 0$). Note, however, that this does *not* rule out the existence of computationally *inefficient* aggregation approaches to construct deterministic predictors. That is, putting computational efficiency aside, it is still an open question whether there exists a sample-efficient and oracle-efficient multi-distribution learning algorithm that outputs a deterministic predictor, and we know from the lower bound of Zhang et al. [23, Theorem 18] that this predictor must be *improper*.

## 2 Hardness of derandomization

In this section, we prove that it is NP-hard to derandomize multi-distribution learning in the most general setup of input distributions $\mathcal{D}_i$ over $\mathcal{X} \times \{-1, 1\}$. In particular, the hardness proof carefully exploits that different data distributions may assign different labels to the same $x \in \mathcal{X}$.

Our NP-hardness proof goes via a reduction from Discrepancy Minimization. In Discrepancy Minimization, we are given as input an $n \times n$ matrix with 0-1 entries. The goal is to find a "coloring" $z \in \{-1, 1\}^n$ such that every entry of $Az$ is as small as possible in absolute value. Formally, we seek to minimize $\|Az\|_\infty$. The seminal work by Charikar et al. [6] showed NP-hardness of computing the best coloring. In full details, their results are as follows.

**Theorem 3** ([6]). *There is a constant $c > 0$ such that it is NP-hard to distinguish whether an input matrix $A \in \{0, 1\}^{n \times n}$ has $\|Az\|_2 \geq cn$ for all $z \in \{-1, 1\}^n$, or whether there exists $z \in \{-1, 1\}^n$ with $Az = 0$.*

Since $\|Az\|_\infty \geq \|Az\|_2/\sqrt{n}$, this similarly implies that it is NP-hard to distinguish whether all $z$ have $\|Az\|_\infty \geq c\sqrt{n}$, or there is a $z$ with $Az = 0$.

Let us now use Theorem 3 to prove our hardness result, Theorem 1. We remark that NP-hardness is formally defined in a uniform model of computation where a Turing Machine takes an encoded input on a tape and decides language membership. As we believe our reduction is clear without going into such formalities, we have deferred a discussion of how to formalize multi-distribution learning in a uniform model of computation to Appendix 4.

*Proof.* Let $n = \min\{d, k/2, c^2/(4\varepsilon^2)\}$ and let $\mathcal{H}$ be an arbitrary hypothesis set of VC-dimension $d$ for which we can find a set of $n$ points that are shattered by $\mathcal{H}$ in $n^{O(1)}$ time. This is possible due to our assumption.

Let $\mathcal{A}$ denote an arbitrary deterministic multi-distribution algorithm. Given a matrix $A \in \{0, 1\}^{n \times n}$ such that either $\|Az\|_\infty \geq c\sqrt{n}$ for all $z \in \{-1, 1\}^n$, or there exists a $z \in \{-1, 1\}^n$ with $Az = 0$, we will now use $\mathcal{A}$ to correctly distinguish these two cases with probability at least $2/3$, thus concluding that the running time of $\mathcal{A}$ is super-polynomial unless $\mathrm{BPP} = \mathrm{NP}$.

Start by computing an arbitrary set $x_1, \ldots, x_n$ of $n$ points that are shattered by $\mathcal{H}$. Now define $2n$ distributions $\mathcal{D}_1^+, \mathcal{D}_1^-, \ldots, \mathcal{D}_n^+, \mathcal{D}_n^-$. Distribution $\mathcal{D}_i^+$ and $\mathcal{D}_i^-$ are both defined from the $i$-th row of $A$. If $m_i$ denotes the number of ones in the $i$-th row of $A$, we let $\mathcal{D}_i^+$ return the sample $(x_j, 1)$ with probability $1/m_i$ for each $j$ with $a_{i,j} = 1$. The distribution $\mathcal{D}_i^-$ similarly returns $(x_j, -1)$ with probability $1/m_i$ for each $j$ with $a_{i,j} = 1$. Observe that these distributions can be described using $n$ bits each.

Now consider running the multi-distribution learning algorithm $\mathcal{A}$ on distributions $\mathcal{D}_1^+, \mathcal{D}_1^-, \ldots, \mathcal{D}_n^+, \mathcal{D}_n^-$ to obtain a deterministic classifier $f : \mathcal{X} \to \{-1, 1\}$. Evaluate $f$ on $x_1, \ldots, x_n$ and compute $\mathrm{er}_{\mathcal{P}}(f)$. This can be done trivially in polynomial time using the definitions of the distributions. If $\mathrm{er}_{\mathcal{P}}(f) < 1/2 + \varepsilon$, then output that there exists $z \in \{-1, 1\}^n$ such that $Az = 0$. Otherwise, output that no such $z$ exists. Clearly this runs in polynomial time. It thus remains to argue correctness.

Consider first the case where there exists $z \in \{-1, 1\}^n$ with $Az = 0$. Since $z$ has inner product $0$ with every row of $A$, it follows that it assigns $1$ to precisely half of the non-zero entries of the $i$-th row and $-1$ to the remaining half. The labeling $z$ of $x_1, \ldots, x_n$ thus has $\mathrm{er}_{\mathcal{D}_i^+}(z) = \mathrm{er}_{\mathcal{D}_i^-}(z) = 1/2$ and $\mathrm{er}_{\mathcal{P}}(z) = 1/2$. Furthermore, since $x_1, \ldots, x_n$ are shattered by $\mathcal{H}$, it follows that $\min_{h \in \mathcal{H}} \mathrm{er}_{\mathcal{P}}(h) \leq 1/2$. By correctness of $\mathcal{A}$, it must hold with probability at least $2/3$ that we correctly output that there exists $z$ with $Az = 0$.

Consider next the case that every $z \in \{-1, 1\}^n$ has $\|Az\|_\infty \geq c\sqrt{n}$. It follows that there is a row $a_i$ such that the vector $v = (f(x_1), \ldots, f(x_n))$ has $|v^T a_i| \geq c\sqrt{n}$. Let $\sigma = \text{sign}(v^T a_i)$. Then

$$
\begin{aligned}
\text{er}_{\mathcal{D}_i^{-\sigma}}(f) &= \frac{1}{m_i} \sum_{j:a_{i,j}=1} \mathbb{1}\{f(x_j) = \sigma\} = \frac{1}{m_i} \sum_{j:a_{i,j}=1} (1/2)(f(x_j)\sigma + 1) \\
&= 1/2 + \frac{1}{2m_i} \sum_{j:a_{i,j}=1} f(x_j)\sigma = 1/2 + \frac{\sigma v^T a_i}{2m_i} \\
&= 1/2 + \frac{|v^T a_i|}{2m_i} \geq 1/2 + c/(2\sqrt{n}).
\end{aligned}
$$

Since we chose $n = \min\{d, k/2, c^2/(4\varepsilon^2)\}$, we have $c/(2\sqrt{n}) \geq \varepsilon$ and thus we return with probability 1 that all $z \in \{-1, 1\}^n$ have $\|Az\|_\infty \geq c\sqrt{n}$. □

Let us end by observing that the distributions used in the above hardness result have OPT $\geq 1/2$. The proof can be modified to prove lower bounds for smaller OPT by adding a dummy point $x_0$ and letting all distributions return $(x_0, 1)$ with probability $1 - 2\text{OPT}'$ and the points in the above distributions with probability $2\text{OPT}'/m_i$. This reduces the value of OPT to around OPT'. However, we also need to reduce $n$ to $\min\{d, k/2, (c \cdot \text{OPT}'/\varepsilon)^2\}$. This agrees well with the fact that for realizable multi-distribution learning, i.e., OPT $= 0$, it is in fact possible to compute a deterministic classifier in polynomial time.

## 3  Deterministic multi-distribution learner

In this section, we give our algorithm for derandomizing multi-distribution learners for label-consistent distributions, i.e., we assume $\mathcal{D}_i(y \mid x) = \mathcal{D}_j(y \mid x)$ for all $i, j, x$.

We start by presenting the high level ideas of our algorithm. Recall that we defined OPT $= \min_{h \in \mathcal{H}} \text{er}_{\mathcal{P}}(h)$, where $\mathcal{P} = \{\mathcal{D}_1, \ldots, \mathcal{D}_k\}$. First, consider running any of the previous randomized multi-distribution learners, producing a distribution $F$ over hypotheses in $\mathcal{H}$ satisfying $\max_i \mathbb{E}_{f \sim F}[\text{er}_{\mathcal{D}_i}(f)] \leq \text{OPT} + \varepsilon/2$. Consider randomly rounding this distribution to a deterministic classifier $\hat{f}$ as follows: For every $x \in \mathcal{X}$ independently (recall that we focus on finite domains), sample an $f \sim F$ and let $\hat{f}(x) = f(x)$. For any distribution $\mathcal{D}_i$, we clearly have $\mathbb{E}_{\hat{f}}[\text{er}_{\mathcal{D}_i}(\hat{f})] = \mathbb{E}_{f \sim F}[\text{er}_{\mathcal{D}_i}(f)] \leq \text{OPT} + \varepsilon/2$. However, as also discussed in the introduction, it is not clear that we can union bound over all $k$ distributions and argue that $\text{er}_{\mathcal{D}_i}(\hat{f}) \leq \text{OPT} + \varepsilon$ for all of them simultaneously. Notice however that the independent choice of $\hat{f}(x)$ for each $x$ gets us most of the way. Indeed, if we let $Z_x$ be a random variable (determined by $\hat{f}(x)$) giving $\text{Pr}_{y \sim \mathcal{D}(y|x)}[\hat{f}(x) \neq y]$, then $\text{er}_{\mathcal{D}_i}(\hat{f}) = \sum_{x \in \mathcal{X}} \mathcal{D}_i(x) Z_x$, where $\mathcal{D}_i(x)$ denotes the probability of $x$ under $\mathcal{D}_i$ and $\mathcal{D}_i(y \mid x)$ gives the conditional distribution of the label $y$ given $x$. Now notice that $\mathcal{D}_i(x) Z_x$ is a random variable taking values in $\{(1/2 - |\beta_x|)\mathcal{D}_i(x), (1/2 + |\beta_x|)\mathcal{D}_i(x)\}$ where $\beta_x = \text{Pr}_{y \sim \mathcal{D}_1(y|x)}[y = 1] - 1/2$ denotes the *bias* of the label of $x$. Furthermore, these random variables are independent. We also have $\mathbb{E}[\text{er}_{\mathcal{D}_i}(\hat{f})] = \mathbb{E}_{f \sim F}[\text{er}_{\mathcal{D}_i}(f)]$. Thus by Hoeffding's inequality

$$
\text{Pr}[|\text{er}_{\mathcal{D}_i}(f) - \mathbb{E}_{f \sim F}[\text{er}_{\mathcal{D}_i}(f)]| > \varepsilon/2] < 2\exp\left(-\frac{\varepsilon^2}{2 \cdot \sum_{x \in X}(2\beta_x \mathcal{D}_i(x))^2}\right).
$$

Examining this expression closely, we observe that this probability is small if $\beta_x \mathcal{D}_i(x)$ is small for all $x \in \mathcal{X}$.

Using this observation, our algorithm then starts by drawing $\tilde{O}(\varepsilon^{-2})$ samples from each distribution $\mathcal{D}_i$ and collecting all $x$ for which the fraction of 1's and $-1$'s is so biased towards either 1 or $-1$, that the majority label almost certainly equals $\text{sign}(\beta_x)$. We then let $\hat{f}(x)$ equal this majority label for all such $x$, and put these $x$ into a set $T$.

What remains is all $x \notin T$. Here we show that these $x$ have so little bias, i.e., $\beta_x \mathcal{D}_i(x)$ is so small, that the random rounding strategy above suffices. The full algorithm is shown as Algorithm 1.

Before giving the formal analysis of the algorithm, note that storing the classifier $\hat{f}$ is quite expensive, as we need to remember the random choice of $\hat{f}(x)$ for every $x \in \mathcal{X} \setminus T$. This is one place where we

---
**Algorithm 1:** DETERMINISTICLEARNER($\mathcal{P}, \varepsilon, \delta, \mathcal{A}$)
---
**Input:** Distributions $\mathcal{P} = \{\mathcal{D}_1, \ldots, \mathcal{D}_k\}$. Precision $\varepsilon > 0$, failure probability $\delta > 0$, randomized
      multi-distribution learner $\mathcal{A}$.

**Result:** Classifier $\hat{f} : \mathcal{X} \to \{-1, 1\}$.

1 Let $C > 0$ be a large enough constant.

2 Let $\gamma = Ck/(\varepsilon\delta)$.

3 Let $T = \emptyset$.

4 Run $\mathcal{A}$ with $\mathcal{P} = \{\mathcal{D}_1, \ldots, \mathcal{D}_k\}$ and $\mathcal{H}$ as input, precision $\varepsilon/2$ and failure probability $\delta/2$ to
  obtain a distribution $F$ over classifiers in $\mathcal{H}$.

5 **for** $i = 1, \ldots, k$ **do**

6      Draw $m = C \ln^2(\gamma)/\varepsilon^2$ samples $\{(x_j, y_j)\}_{j=1}^m$ from $\mathcal{D}_i$.

7      For every $x \in \mathcal{X} \setminus T$ such that $n_{i,x} := |\{j : x_j = x\}| > 0$, let
     $\rho_{i,x} = (|\{j : x_j = x \wedge y_j = 1\}| - |\{j : x_j = x \wedge y_j = -1\}|)/n_{i,x}$.

8      For every $x \in \mathcal{X} \setminus T$ such that $n_{i,x} > 0$, if $|\rho_{i,x}| > \sqrt{\ln(\gamma)/n_{i,x}}$, add $x$ to $T$ and let
     $\hat{f}(x) = \text{sign}(\rho_{i,x})$.

9 For every $x \in \mathcal{X} \setminus T$, independently draw an $f \sim F$ and let $\hat{f}(x) = f(x)$.

10 **return** $\hat{f}$
---

use the assumption that $\mathcal{X}$ is finite. Note however that even for finite $\mathcal{X}$, storing $|\mathcal{X}|$ random choices to represent the classifier might be infeasible. Furthermore, the sampling of $\hat{f}(x)$ for every $x$ also adds $|\mathcal{X}|$ to the running time, which is again too expensive. We propose a method for reducing the storage and running time requirement later in this section. For now, we analyse Algorithm 1 without worrying about $|\mathcal{X}|$.

**Analysis.** In our analysis, we separately handle $x \in T$ and $x \notin T$. The two technical results we need are stated next. First, define the *bias* $\beta_x$ of an $x \in \mathcal{X}$ as $\Pr_{y \sim \mathcal{D}_1(y|x)}[y = 1] - 1/2$. We say that an $x$ is *heavily biased* if

$$\beta_x^2 \mathcal{D}_i(x) > \frac{\varepsilon^2}{8 \cdot \ln(4k/\delta)}$$

for at least one $i$, and *lightly biased* otherwise. Intuitively, our algorithm ensures that $T$ contains all heavily biased $x$ and that all predictions made on $x \in T$ are correct. This is stated in the following

**Lemma 4.** *It holds with probability at least $1 - \delta/4$ that every heavily biased $x$ is in $T$, and furthermore, for every $x \in T$, we have $\hat{f}(x) = \text{sign}(\beta_x)$.*

Next, we also show that random rounding outside $T$ suffices.

**Lemma 5.** *Assume every heavily biased $x$ is in $T$ after the for-loop. Then with probability at least $1 - \delta/4$ over the random choice of $\hat{f}(x)$ with $x \in \mathcal{X} \setminus T$, it holds for all $i$ that*

$$\left| \mathbb{E}_{f \sim F}[\mathbb{E}_{(x,y) \sim \mathcal{D}_i}[1\{x \notin T \wedge f(x) \neq y\}]] - \mathbb{E}_{(x,y) \sim \mathcal{D}_i}[1\{x \notin T \wedge \hat{f}(x) \neq y\}] \right| \leq \varepsilon/2.$$

Before giving the proof of Lemma 4 and Lemma 5, let us use these two results to complete the proof of Theorem 2.

*Proof of Theorem 2.* From a union bound and Lemma 4 and Lemma 5, we have with probability $1 - \delta$, that all of the following hold

- The invocation of $\mathcal{A}$ in step 1 of Algorithm 1 returns a distribution $F$ with $\max_i \mathbb{E}_{f \sim F}[\text{er}_{\mathcal{D}_i}(f)] \leq \text{OPT} + \varepsilon/2$.

- For every $x \in T$, we have $\hat{f}(x) = \text{sign}(\beta_x)$.

- For every distribution $\mathcal{D}_i$,

$$\left| \mathbb{E}_{f \sim F}[\mathbb{E}_{(x,y) \sim \mathcal{D}_i}[1\{x \notin T \wedge f(x) \neq y\}]] - \mathbb{E}_{(x,y) \sim \mathcal{D}_i}[1\{x \notin T \wedge \hat{f}(x) \neq y\}] \right| \leq \varepsilon/2.$$

Assume now that all of the above hold. We rewrite $\mathrm{er}_{\mathcal{P}}(\hat{f})$ by splitting the contributions to the error into $x \in T$ and $x \notin T$,

$$
\begin{aligned}
\mathrm{er}_{\mathcal{P}}(\hat{f}) &= \max_i \mathrm{er}_{\mathcal{D}_i}(\hat{f}) \\
&= \max_i \left( \mathbb{E}_{(x,y)\sim\mathcal{D}_i}[1\{x \in T \wedge \hat{f}(x) \neq y\}] + \mathbb{E}_{(x,y)\sim\mathcal{D}_i}[1\{x \notin T \wedge \hat{f}(x) \neq y\}] \right).
\end{aligned}
$$

Using that $\hat{f}(x) = \mathrm{sign}(\beta_x)$ for $x \in T$, we have $\mathbb{E}_{(x,y)\sim\mathcal{D}_i}[1\{x \in T \wedge \hat{f}(x) \neq y\}] = \min_{z\in\{-1,1\}^T}[\mathbb{E}_{\mathcal{D}_i}[1\{x \in T \wedge z(x) \neq y\}]$. Thus the above is bounded by

$$
\max_i \left( \min_{z\in\{-1,1\}^T}[\mathbb{E}_{\mathcal{D}_i}[1\{x \in T \wedge z(x) \neq y\}] + \mathbb{E}_{f\sim F}[\mathbb{E}_{\mathcal{D}_i}[1\{x \notin T \wedge f(x) \neq y\}]] \right) + \varepsilon/2.
$$

Since every $f$ in the support of $F$ is a deterministic classifier, we have

$$
\min_{z\in\{-1,1\}^T}[\mathbb{E}_{\mathcal{D}_i}[1\{x \in T \wedge z(x) \neq y\}] \leq \mathbb{E}_{f\sim F}[\mathbb{E}_{\mathcal{D}_i}[1\{x \in T \wedge f(x) \neq y\}].
$$

We therefore have

$$
\begin{aligned}
\mathrm{er}_{\mathcal{P}}(\hat{f}) &\leq \max_i \left( \mathbb{E}_{f\sim F}[\mathbb{E}_{\mathcal{D}_i}[1\{x \in T \wedge f(x) \neq y\}]] + \mathbb{E}_{f\sim F}[\mathbb{E}_{\mathcal{D}_i}[1\{x \notin T \wedge f(x) \neq y\}]] \right) + \varepsilon/2 \\
&= \max_i \mathbb{E}_{f\sim F}[\mathrm{er}_{\mathcal{D}_i}(f)] + \varepsilon/2 \\
&\leq \mathrm{OPT} + \varepsilon.
\end{aligned}
$$

This completes the proof of Theorem 2. $\qquad\square$

*Proof of Lemma 4.* We first define the two types of failures that may occur:

- For every $i$ and every $x$ with $\beta_x^2 \mathcal{D}_i(x) > \varepsilon^2/(8\ln(4k/\delta))$, let $E_{i,x}^1$ denote the event that $n_{i,x} < (C/2)\beta_x^{-2}\ln\gamma$.

- For every $i$, let $E_i^2$ denote the event that there is an $x$ with $n_{i,x} > 0$ and $|\beta_x - \rho_{i,x}/2| > \sqrt{\ln(\gamma)/(16n_{i,x})}$.

Assume first that none of the events occur. Consider a heavily biased $x$. Then there is an $i$ for which $\beta_x^2 \mathcal{D}_i(x) > \varepsilon^2/(8\ln(4k/\delta))$. Since $E_{i,x}^1$ does not occur, we have $n_{i,x} \geq (C/2)\beta_x^{-2}\ln\gamma$. Since $E_i^2$ does not occur, we also have $|\beta_x - \rho_{i,x}/2| \leq \sqrt{\ln(\gamma)/(16n_{i,x})}$. Hence $|\rho_{i,x}| \geq 2|\beta_x| - 2\sqrt{\ln(\gamma)/(16n_{i,x})}$. But $|\beta_x| \geq \sqrt{(C/2)\ln(\gamma)/n_{i,x}}$ and thus $|\rho_{i,x}| \geq (\sqrt{2C} - 1/2)\sqrt{\ln(\gamma)/n_{i,x}}$. For $C$ large enough, this is at least $\sqrt{\ln(\gamma)/n_{i,x}}$, which puts $x$ in $T$ during step 8 of Algorithm 1. Thus every heavily biased $x$ is in $T$. Secondly, note that when an $x$ is added to $T$ in iteration $i$ of the for-loop, we have $|\rho_{i,x}| > \sqrt{\ln(\gamma)/n_{i,x}}$. Since $E_i^2$ does not occur, we have $|\beta_x - \rho_{i,x}/2| \leq \sqrt{\ln(\gamma)/(16n_{i,x})}$. But this implies $\beta_x \in [\rho_{i,x}/2 - \sqrt{\ln(\gamma)/(16n_{i,x})}, \rho_{i,x}/2 + \sqrt{\ln(\gamma)/(16n_{i,x})}]$. Since $|\rho_{i,x}| > \sqrt{\ln(\gamma)/n_{i,x}}$, every number in this interval has the same sign as $\rho_{i,x}$, i.e. $\hat{f}(x) = \mathrm{sign}(\rho_{i,x}) = \mathrm{sign}(\beta_x)$. Thus what remains is to bound the probability of these events.

For $E_{i,x}^1$, fix an $i$ and $x$ with $\beta_x^2 \mathcal{D}_i(x) > \varepsilon^2/(8\ln(4k/\delta))$, we have

$$
\mathbb{E}[n_{i,x}] = \mathcal{D}_i(x)m > \varepsilon^2 m/(8\beta_x^2 \ln(4k/\delta)) > C\beta_x^{-2}\ln\gamma.
$$

For $C$ large enough, we get from a Chernoff bound that $\Pr[E_{i,x}^1] = \Pr[n_{i,x} < (C/2)\beta_x^{-2}\ln\gamma] < \gamma^{-2}$.

For $E_i^2$, let us first condition on an outcome of the values $n_{i,x}$ for all $x$. Then for every $x$, we have that $p_{i,x} := |\{j : x_j = x \wedge y_j = 1\}| - |\{j : x_j = x \wedge y_j = -1\}|$ is distributed as the sum of $n_{i,x}$ independent $-1/1$ random variables taking the value 1 with probability $\beta_x + 1/2$. Hence $\mathbb{E}[p_{i,x}] = 2\beta_x n_{i,x}$. Since $\rho_{i,x} = p_{i,x}/n_{i,x}$, it follows from Hoeffding's inequality that

$$
\begin{aligned}
\Pr\left[|\beta_x - \rho_{i,x}/2| > \sqrt{\ln(\gamma)/n_{i,x}}\right] &= \Pr\left[|2\beta_x n_{i,x} - p_{i,x}| > 2\sqrt{\ln(\gamma)n_{i,x}}\right] \\
&< 2\exp\left(-\frac{8\ln(\gamma)n_{i,x}}{4n_{i,x}}\right) = 2\gamma^{-2}.
\end{aligned}
$$

For any fixed values $n_{i,x}$, there are at most $m$ distinct $x$ with a non-zero $n_{i,x}$. A union bound over all of them implies $\Pr[E_i^2 \mid n_{i,x}] \leq 2m\gamma^{-2}$. Since this upper bound holds for any outcome of the $n_{i,x}$, we have also $\Pr[E_i^2] \leq 2m\gamma^{-2}$.

We now observe that for every $i$, there are at most $\varepsilon^{-2}8\ln(4k/\delta)$ distinct $x$ with $\beta_x^2 \mathcal{D}_i(x) > \varepsilon^2/(8\ln(4k/\delta))$. Hence $\Pr[\cup_x E_{i,x}^1] \leq \varepsilon^{-2}8\ln(4k/\delta)\gamma^{-2}$. A union bound over all $i$ finally implies

$$\Pr[(\cup_i \cup_x E_{i,x}^1) \cup (\cup_i E_i^2)] \leq k\gamma^{-2}\left(\varepsilon^{-2}8\ln(4k/\delta) + 2m\right)$$

Since $\gamma = Ck/(\varepsilon\delta)$ and $m = C\ln^2(\gamma)/\varepsilon^2$, we have for large enough $C$ that this probability is bounded by $\delta/4$. $\qquad\square$

*Proof of Lemma 5.* Fix a distribution $\mathcal{D}_i$. Observe that for any $x \in \mathcal{X} \setminus T$, we have that the distribution of $\hat{f}(x)$ is the same as $f(x)$ for $f \sim F$. Hence $\mathbb{E}_{\hat{f}}[\mathbb{E}_{(x,y)\sim\mathcal{D}_i}[1\{x \notin T \wedge \hat{f}(x) \neq y\}]] = \mathbb{E}_{f\sim F}[\mathbb{E}_{(x,y)\sim\mathcal{D}_i}[1\{x \notin T \wedge f(x) \neq y\}]]$. Denote this expectation by $\mu$. If we let $Z_x$ be the random variable (as a function of $\hat{f}(x)$) taking the value $\Pr_{y\sim\mathcal{D}_i(y|x)}[\hat{f}(x) \neq y]$, then

$$\mathbb{E}_{(x,y)\sim\mathcal{D}_i}[1\{x \notin T \wedge \hat{f}(x) \neq y\}] = \sum_{x\in\mathcal{X}\setminus T} \mathcal{D}_i(x)Z_x.$$

Observe that $Z_x$ is either $1/2 - |\beta_x|$ or $1/2 + |\beta_x|$, depending on whether $\hat{f}(x) = \text{sign}(\beta_x)$ or not. Hence $\mathcal{D}_i(x)Z_x \in [\mathcal{D}_i(x)(1/2 - |\beta_x|), \mathcal{D}_i(x)(1/2 + |\beta_x|)]$ and the $Z_x$ are independent. We thus get from Hoeffding's inequality and that $x \notin T$ are lightly biased that

$$\Pr_{\hat{f}}\left[\sum_{x\in\mathcal{X}\setminus T} \mathcal{D}_i(x)Z_x > \mu + \varepsilon/4\right] < \exp\left(\frac{-2(\varepsilon/2)^2}{\sum_{x\in\mathcal{X}\setminus T}(2|\beta_x|\mathcal{D}_i(x))^2}\right)$$

$$\leq \exp\left(\frac{-\varepsilon^2}{\sum_{x\in\mathcal{X}\setminus T}\mathcal{D}_i(x)\varepsilon^2/\ln(4k/\delta)}\right)$$

$$\leq \exp\left(-\ln(4k/\delta)\right) = \delta/(4k).$$

A union bound over all $\mathcal{D}_i$ completes the proof. $\qquad\square$

## 3.1 Reducing storage and time

The above description of Algorithm 1 requires the storage of an independent random choice of $\hat{f}(x)$ for every $x \in \mathcal{X}$. This is infeasible for large $\mathcal{X}$, both in terms of space usage and the time needed for making these random choices. Instead, we can reduce the storage requirements by using an $r$-wise independent hash function $q : \mathcal{X} \to \mathcal{Y}$ for a sufficiently large output domain $\mathcal{Y}$ to make the random rounding. Recall that an $r$-wise independent hash function hashes any set of up to $r$ distinct keys $x_1, \ldots, x_r$ independently and uniformly at random into $\mathcal{Y}$. Such a hash function can be implemented in space $O(r\ln(|\mathcal{X}||\mathcal{Y}|))$ bits and evaluated in time $\tilde{O}(r\ln(|\mathcal{X}||\mathcal{Y}|))$ by e.g., interpreting an $x \in \mathcal{X}$ as an index into $[|\mathcal{X}|] = \{0, \ldots, |\mathcal{X}| - 1\}$ and letting $q(x) = \sum_{i=0}^{r-1} \alpha_i x^i \pmod{p}$ for a prime $p = |\mathcal{Y}| > |\mathcal{X}|$ and the $\alpha_i$ independent and uniformly random in $[p]$. Using fast multiplication algorithms, $q(x)$ can be evaluated in time $\tilde{O}(r\ln(|\mathcal{X}||\mathcal{Y}|))$, even when $\ln(|\mathcal{X}||\mathcal{Y}|)$ bits does not fit in a machine word. The time to sample the hash function is only $O(r\ln|\mathcal{X}||\mathcal{Y}|)$ (we just need the random coefficients of the polynomial).

Instead of storing $\hat{f}(x)$ for every $x \in \mathcal{X} \setminus T$ explicitly, the learning algorithm instead stores $q$ and the distribution $F$. Given this information, we evaluate $\hat{f}(x)$ by computing $q(x)$ and letting $\hat{f}(x) = 1$ if $q(x) \leq \Pr_{f\sim F}[f(x) = 1]|\mathcal{Y}| - 1$ and $-1$ otherwise. Since $q(x)$ is uniform over $\mathcal{Y}$ for any $x$, we have $\Pr[\hat{f}(x) = 1] = \lfloor \Pr_{f\sim F}[f(x) = 1]|\mathcal{Y}|\rfloor/|\mathcal{Y}|$. This probability satisfies $\Pr_{f\sim F}[f(x) = 1] - 1/|\mathcal{Y}| \leq \Pr[\hat{f}(x) = 1] \leq \Pr_{f\sim F}[f(x) = 1]$ and is thus almost the same rounding probability as in Algorithm 1. Since previous multi-distribution learning algorithms also store $F$, this only adds $O(r\ln(|\mathcal{X}||\mathcal{Y}|))$ bits to the storage.

What remains is to determine an $r$ and $|\mathcal{Y}|$ for which this is sufficient for the guarantees of Algorithm 1. We will show that $r = 2\ln(4k/\delta)$ and $|\mathcal{Y}| = \Theta(\varepsilon^{-3}\ln(k/\delta))$ suffices if we increase the sample

complexity of Algorithm 1 by a logarithmic factor. Observe that the $O(r \ln(|\mathcal{X}| \ln(k/\delta)/\varepsilon))$ extra bits is only proportional to storing $O(\ln(k/\delta))$ samples from $\mathcal{X}$, provided that $\ln(k/\delta)/\varepsilon$ is no larger than a polynomial in $|\mathcal{X}|$. The space overhead is thus very minor.

We only give an outline of how to modify the proof in the previous section to work with $r$-wise independence as it follows the previous proof rather uneventfully. First, redefine the threshold for being heavily biased to $\beta_x^2 \mathcal{D}_i(x) > \varepsilon^2/(C' \ln^2(4k/\delta))$ for large enough constant $C'$.

For the proof of Lemma 4 to still go through, this requires us to increase $m$ by a $C' \ln \gamma$ factor, i.e. to $CC' \ln^3(\gamma)/\varepsilon^2$, and also increase $\gamma$ by $C'$ to $CC'k/(\varepsilon\delta)$. Then the only change to the proof, is that we have an event $E_{i,x}^1$ for every $i$ and every $x$ with $\beta_x^2 \mathcal{D}_i(x) > \varepsilon^2/(C' \ln^2(4k/\delta))$. Otherwise, all conditions in the events $E_{i,x}^1$ and $E_i^2$ remain the same. Thus the proof still goes through if we can argue $\Pr[E_{i,x}^1] \leq \gamma^{-2}$. So fix an $i$ and $x$ with $\beta_x^2 \mathcal{D}_i(x) > \varepsilon^2/(C' \ln^2(4k/\delta))$. Then $\mathbb{E}[n_{i,x}] = \mathcal{D}_i(x)m > \varepsilon^2 m/(C' \beta_x^2 \ln^2(4k/\delta)) > C\beta_x^{-2} \ln \gamma$. This is the same lower bound on $\mathbb{E}[n_{i,x}]$ as the previous proof and thus we can complete the steps. Finally, note that we finished the proof of Lemma 4 by a union bound. Here we needed $k\gamma^{-2}(\varepsilon^{-2}8\ln(4k/\delta) + 2m) < \delta/4$. This is still the case for our new $m$ and $\gamma$.

Now for the proof of Lemma 5, we used Hoeffding's inequality. This requires the random rounding to be independent for different $x$. With our modified approach, the roundings are only $r$-wise independent and thus we need the following variant of Hoeffding's inequality for $r$-wise independent random variables

**Theorem 6** ([17]). *Let $Z_1, \ldots, Z_n$ be a sequence of $r$-wise independent random variables for $r \geq 2$ with $|Z_i - \mathbb{E}[Z_i]| \leq 1$ for all outcomes. Let $Z = \sum_i Z_i$ with $\mathbb{E}[Z] = \mu$ and let $\sigma^2(Z) = \sum_i \sigma^2(Z_i)$ denote the variance of $Z$. Then the following holds for even $r$ and any $Q \geq \max\{r, \sigma^2(Z)\}$:*

$$\Pr[|Z - \mu| \geq T] \leq \left(\frac{rQ}{e^{2/3}T^2}\right)^{r/2}.$$

If we repeat the proof of Lemma 5, define $Z_x$ as the random variable (as a function of the random choice of $q$) taking the value $\Pr_{y \sim \mathcal{D}_i(y|x)}[\hat{f}(x) \neq y]$. Note that $Z_x \in \{1/2 - |\beta_x|, 1/2 + |\beta_x|\}$. This also implies that $|Z_x - \mathbb{E}[Z_x]| \leq 2|\beta_x|$ for all outcomes of $Z_x$. When all heavily biased $x$ are in $T$, we have $\beta_x^2 \mathcal{D}_i(x) \leq \varepsilon^2/(C' \ln^2(4k/\delta))$ for all $x \notin T$. This implies $|\beta_x| \leq \varepsilon/(\ln(4k/\delta)\sqrt{C'\mathcal{D}_i(x)})$. Now let $\alpha = 2\varepsilon/(\ln(4k/\delta)\sqrt{C'})$. Then

$$\mathbb{E}_{(x,y) \sim \mathcal{D}_i}[1\{x \notin T \wedge \hat{f}(x) \neq y\}] = \sum_{x \in \mathcal{X} \setminus T} \mathcal{D}_i(x)Z_x = \alpha \sum_{x \in \mathcal{X} \setminus T} \frac{\mathcal{D}_i(x)Z_x}{\alpha}.$$

The random variable $\mathcal{D}_i(x)Z_x/\alpha$ thus satisfies $|\mathcal{D}_i(x)Z_x/\alpha - \mathbb{E}[\mathcal{D}_i(x)Z_x/\alpha]| \leq 2\mathcal{D}_i(x)|\beta_x|/\alpha \leq \sqrt{\mathcal{D}_i(x)} \leq 1$ for all outcomes. This also gives us $\sigma^2(\mathcal{D}_i(x)Z_x/\alpha) \leq \mathcal{D}_i(x)$ and thus

$$\sigma^2\left(\sum_{x \in \mathcal{X} \setminus T} \frac{\mathcal{D}_i(x)Z_x}{\alpha}\right) \leq \sum_{x \in \mathcal{X} \setminus T} \mathcal{D}_i(x) \leq 1.$$

Now consider the expected value (with $a \pm b = [a - b, a + b]$)

$$
\begin{aligned}
\mu' &= \mathbb{E}[\sum_{x \in \mathcal{X} \setminus T} \mathcal{D}_i(x)Z_x] \\
&= \sum_{x \in \mathcal{X} \setminus T} \mathcal{D}_i(x)\mathbb{E}_q[\Pr_{y \sim \mathcal{D}_i(y|x)}[\hat{f}(x) \neq y]] \\
&\in \sum_{x \in \mathcal{X} \setminus T} \mathcal{D}_i(x)\left(\mathbb{E}_{f \sim F}[\Pr_{y \sim \mathcal{D}_i(y|x)}[f(x) \neq y]] \pm 1/|\mathcal{Y}|\right) \\
&\subseteq \mathbb{E}_{f \sim F}[\mathbb{E}_{(x,y) \sim \mathcal{D}_i}[1\{x \notin T \wedge f(x) \neq y\}]] \pm 1/|\mathcal{Y}|.
\end{aligned}
$$

Letting $\mu = \mathbb{E}_{f \sim F}[\mathbb{E}_{(x,y) \sim \mathcal{D}_i}[1\{x \notin T \wedge f(x) \neq y\}]]$, we then have by Theorem 6 with $Q = r$ that

$$\Pr\left[\left|\sum_{x \in \mathcal{X} \setminus T} \mathcal{D}_i(x)Z_x - \mu\right| \geq \alpha T\right] \leq \Pr\left[\left|\sum_{x \in \mathcal{X} \setminus T} \mathcal{D}_i(x)Z_x - \mu'\right| \geq \alpha T - 1/|\mathcal{Y}|\right]$$

$$= \Pr\left[\left|\sum_{x \in \mathcal{X} \setminus T} \frac{\mathcal{D}_i(x)Z_x}{\alpha} - \mu'/\alpha\right| \geq T - \alpha/|\mathcal{Y}|\right]$$

$$\leq \left(\frac{r^2}{e^{2/3}(T - \alpha/|\mathcal{Y}|)^2}\right)^{r/2}.$$

Inserting $T = \varepsilon/(2\alpha)$ and using $r = 2\ln(4k/\delta)$, $|\mathcal{Y}| \geq 4\alpha^2/\varepsilon$ gives $(T - \alpha/|\mathcal{Y}|) \geq \varepsilon/(4\alpha)$ and thus finally implies

$$\Pr\left[\left|\mathbb{E}_{(x,y) \sim \mathcal{D}_i}[1\{x \notin T \wedge \hat{f}(x) \neq y\}] - \mathbb{E}_{f \sim F}[\mathbb{E}_{(x,y) \sim \mathcal{D}_i}[1\{x \notin T \wedge f(x) \neq y\}]]\right| \geq \varepsilon/2\right]$$

$$\leq \left(\frac{16r^2\alpha^2}{e^{2/3}\varepsilon^2}\right)^{r/2} = \left(\frac{64r^2}{C'e^{2/3}\ln^2(4k/\delta)}\right)^{r/2} = \left(\frac{256}{C'e^{2/3}}\right)^{r/2}$$

$$\leq e^{-r/2} = \delta/(4k).$$

Here, the last inequality follows for $C'$ large enough. Thus, if we increase the sample complexity to $m(k, d, \mathrm{OPT}, \varepsilon/2, \delta/2) + O(k\ln^3(k/(\varepsilon\delta))/\varepsilon^2)$, then we may sample and store a hash function using only $O(\ln(n/\delta)\ln(|\mathcal{X}|\ln(k/\delta)/\varepsilon))$ extra bits and $O(\ln(n/\delta)\ln(|\mathcal{X}|\ln(k/\delta)/\varepsilon))$ time.

## 3.2 Infinite Input Domains

In the above presentation of our algorithm, we have assumed a finite input domain $\mathcal{X}$. While we believe this is a very reasonable assumption, we here present some initial ideas for how this restrictions might be circumvented.

Assume that the black-box randomized multi-distribution learner $\mathcal{A}$ always outputs a distribution $F$ over a finite number of classifiers in $\mathcal{H}$. Let $m$ be an upper bound on the size of the support. Then since $\mathcal{H}$ has VC-dimension $d$, the *dual* VC-dimension is at most $2^d$ [1]. By Sauer-Shelah, this implies that the number of distinct ways $x \in \mathcal{X}$ may be labeled by the support of $F$ is bounded $\binom{m}{2^d+1}$, i.e. finite. We believe that treating just the distinct ways $x$ is labeled by the hypotheses in the support should be sufficient to recover our results for finite $\mathcal{X}$.

## Acknowledgments

Kasper Green Larsen is co-funded by a DFF Sapere Aude Research Leader Grant No. 9064-00068B by the Independent Research Fund Denmark and co-funded by the European Union (ERC, TUCLA, 101125203). Views and opinions expressed are however those of the author(s) only and do not necessarily reflect those of the European Union or the European Research Council. Neither the European Union nor the granting authority can be held responsible for them. Omar Montasser was supported by a FODSI-Simons postdoctoral fellowship at UC Berkeley.

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

## 4    Uniform Model of Computation

For a fully formalized NP-hardness proof, we technically need to define an input encoding of a multi-distribution learning problem and argue that the sampling steps may be simulated by a Turing Machine. Furthermore, details such as whether the hypothesis set $\mathcal{H}$ is part of the input or known to the algorithm also needs to be formalized. In this section, we discuss various choices one could make. We note that similar discussions and formalizations of learning in a uniform model of computation has been carefully carried out in classic learning theory books [11].

First, we find it most natural that $\mathcal{H}$ is part of the learning problem, i.e. not an input to the algorithm, but is allowed to be "hard-coded" into the algorithm. This is the best match to standard learning problems, where e.g. the Support Vector Machine learning algorithm, or Logistic Regression via gradient descent, knows that we are working with linear models. Similarly, the input domain seems best modeled by letting it be known to the algorithm. One tweak could be that if the input is $d$-dimensional vectors, then $d$ could be part of the input to the algorithm. This again matches how most natural learning algorithms work for arbitrary $d$ (and our proof needs $d$ to grow for our $n$ to grow).

Now regarding modeling multi-distribution learning, we find that the following uniform computational model most accurately matches what the community thinks of as multi-distribution learning (here stated for the input domain being $n$-dimensional vectors and the hypothesis set being linear models).

A solution to multi-distribution learning with linear models, is a special Turing machine $M$. $M$ receives as input a number $n$ on the input tape. In addition to a standard input/output tape and a tape with random bits, $M$ has a "sample"-tape, a "target distribution"-tape and a special "sample"-state. When $M$ enters the "sample"-state, the bits on the "target distribution" tape is interpreted as an index in $i$ and the contents of the "sample"-tape is replaced by a binary description of a fresh sample from a distribution $\mathcal{D}_i$ ($\mathcal{D}_i$ is only accessible through the "sample"-state). A natural assumption here would be that $\mathcal{D}_i$ is only supported over points with integer coordinates bounded by $n$ in magnitude. This gives a natural binary representation of each sample using $n \log n$ bits, plus one bit for the label.

$M$ runs until terminating in a special halt state, with the promise that regardless of what $n$ distributions $\mathcal{D}_1, \ldots, \mathcal{D}_n$ over the input domain that are used for generating samples in the "sample"-state, it holds with probability at least $2/3$ over the samples and the random bits on the tape, that the output tape contains a binary encoding of a hyperplane with error at most $\tau + 1/n$ for every distribution $\mathcal{D}_i$. A bit more generally, we could also let it terminate with an encoding of a Turing machine on its output tape. That Turing machine, upon receiving the encoding of $n$ and an $n$-dimensional point on its input tape, outputs a prediction on its tape. This allows more general hypotheses than just outputting something from $\mathcal{H}$.

The above special states and tapes are introduced to most accurately represent multi-distribution learning. Now observe that our reduction from discrepancy minimization still goes through. Given

such a special Turing machine $M$ for multi-distribution learning, observe that we can obtain a standard (randomized) Turing machine $M'$ for discrepancy minimization from it. Concretely, in discrepancy minimization, the input is the integer $n$ and an $n \times n$ binary matrix $A$. As mentioned in our reduction, we can easily compute $n$ shattered points for linear models, e.g. just the standard basis $e_1, \ldots, e_n$. Now do as in our reduction and interpret each row of $A$ as two distributions over $e_1, \ldots, e_n$. $M'$ can now simulate the "sample"-state, "sample"-tape and "target distribution" tape of $M$, as it can itself use its random tape to generate samples from the distributions. In this way, $M'$ can simulate $M$ without the need for special tapes and states, and by the guarantees of $M$ (as in our reduction), it can distinguish whether $A$ has discrepancy 0 or $\Omega(\sqrt{n})$ by using the final output hypothesis of $M$ and evaluating it on $e_1, \ldots, e_n$ and computing the error on each of the (known) distributions $\mathcal{D}_i$ obtained from the input matrix $A$.

Note that the reduction would also hold if we rephrased multi-distribution learning such that the algorithm receives some binary encoding of $\mathcal{D}_1, \ldots, \mathcal{D}_n$ as input. This would make the reduction even more straight-forward, as we need not worry about samples. However, we feel the above definition with a special state and tapes for sampling more accurately represent multi-distribution learning from a learning theoretic perspective. We thus prefer a slightly more complicated reduction as above to better model the problem.

