# OpenReview forum: "Derandomizing Multi-Distribution Learning"
_NeurIPS.cc/2024/Conference — NeurIPS 2024 poster_

### Official Review · Reviewer_x5BR · 2024-07-08

**Soundness:** 3
**Presentation:** 2
**Contribution:** 3
**Rating:** 7
**Confidence:** 3

**Summary:**

Multi-distribution learning is an extension of the agnostic learning setup which defined as follows. Given a collection of k distributions $(D_1, …, D_k)$ over the domain $\mathcal{X} \times \\{\pm 1\\}$ and baseline hypothesis class $\mathcal{H}$ find a (potentially randomized) classifier $f: \mathcal{X} \to \\{\pm 1\\}$ such that its *worst-case* loss $\max_{i \in [k]} \mathbb{E}_{f \sim \mathcal{F}}L_i(f)$ is competitive with that of the best fixed classifier in $\mathcal{H}$. Here, $L_i$ denotes the zero-one loss $L_i(f) = \Pr_i[f(X) \neq Y]$ where the probability $\Pr_i$ defined is over $(X, Y) \sim D_i$.

Previous work has shown that one can construct *randomized* classifiers (i.e., a distribution $\mathcal{F}$ over classifiers) that succeeds in multi-distribution learning with sample complexity proportional to the VC dimension of the baseline class $\mathcal{H}$. This work addresses the question of whether it is possible to efficiently "derandomize" such classifiers to achieve multi-distribution learning with a *deterministic* classifier. The main contributions are twofold.

1. Learning a deterministic classifier in the multi-distribution learning setting is *computationally* hard since **discrepancy minimization**, which is an NP-hard problem, reduces to multi-distribution learning.
2. If we assume that the given distributions $(D_1,...,D_k)$ share a common conditional distribution $f^*(x) = \Pr[Y = 1|X=x]$, then derandomization is possible by "essentially" (see comment below) labeling *each* $x \in \mathcal{X}$ with independently drawn classifiers $f \sim \mathcal{F}$.

"essentially": the statement is true for marginal distributions with high min-entropy. For distributions with "heavy" atoms, there is a simple fix; if $x$ is a heavy element, estimate the conditional probability $f^*(x) = \Pr[Y=1|X=x]$ with sufficient confidence (which is possible since $x$ is a heavy element) and label it with $\mathrm{sign}(f^*(x))$.

**Strengths:**

The main results of this work stem from simple but powerful observations. The insistence on learning *deterministic* classifiers turns the multi-distribution learning problem into a combinatorial one, which already suggests that some NP-hardness result could be right around the corner. The authors drive this intuition home by connecting the multi-distribution learning to discrepancy minimization, which is known to be NP-hard. This connection, though simple to establish, appears to be fundamental and may lead to exciting opportunities for cross-pollination between learning theory and discrepancy theory. Below is a technical description highlighting the key insight of the main reduction.

**Reduction from discrepancy minimization.** Discrepancy minimization is defined as follows. Let $c > 0$ be any fixed constant. Given an $n \times n$ matrix $A \in \\{0,1\\}^{n \times n}$ distinguish between the case where 1) there exists $z \in \\{\pm 1\\}^n$ such that $Az = \mathbf{0}$ and 2) $||Az||_\infty \ge c \sqrt{n}$ for all $z \in \\{\pm 1\\}^n$. For some constant $c > 0$, this problem is NP-hard.

The authors' key insight is the following. Consider two "antithetical" distributions $D_1, D_{-1}$ where $D_1, D_{-1}$ share the same marginal distribution $\mu_1$ supported on an $n$-point set $\\{x_1,...,x_n\\}$ but $D_1$ assigns all the mass on the set $\\{(x,1) | x \in \\{x_1,...,x_n\\}\\}$ and $D_{-1}$ only on $\\{(x,-1) | x \in \\{x_1,...,x_n\\}\\}$. Then, the worst-case zero-one loss achievable w.r.t. $D_1,D_{-1}$ by *any* classifier is 1/2. Moreover, any *deterministic* classifier that achieves the optimal value OPT = 1/2 "perfectly balances" the $n$ points $x_1,...,x_n$ w.r.t. the marginal measure $\mu_1$. Let $A_i \in \\{0,1\\}^n$ be the $i$-th row of the discrepancy instance $A \in \\{0,1\\}^{n \times n}$. If there exists $z \in \\{\pm 1\\}^n$ such that $A z = \mathbf{0}$, then $A_i z = 0$. We can convert each row $A_i$ into a pair of distributions $D_{i}, D_{-i}$ such that OPT=1/2 if and only if there exists $z \in \\{\pm 1\\}^n$ s.t. $Az = \mathbf{0}$.

Note the minimal role of the baseline class $\mathcal{H}$. All we require is that $\mathcal{H}$ shatter the $n$-point support $\{x_1,...,x_n\}$. A sufficient condition that satisfies this is $\mathrm{VCdim}(\mathcal{H}) > n$.

**Weaknesses:**

While this work constitutes a solid contribution to learning theory, it falls a little short in the following aspects.

**Formal definition of a mutli-distribution learning in the uniform computational model?** While the key idea behind the reduction from discrepancy minimization is clear, some formalities in making the reduction rigorous seems to be missing. NP-hardness is a notion defined for uniform computational models (e.g., Turing machines). To use NP-hardness of discrepancy minimization as the basis for the hardness of multi-distribution learning, one needs to define what it means to solve multi-distribution learning using Turing machines. Suppose we want to say a Turing machine M "solves" multi-distribution learning. What are the inputs to M? Is it the collection of distributions $(D_1,...,D_k)$? Does the input also include the baseline class $\mathcal{H}$? How are these inputs represented?

In addition, computational complexity is an asymptotic notion which requires a *sequence* of problems, not just a single instance. For example, discrepancy minimization instances are binary matrices $A \in \\{0,1\\}^{n \times n}$, which can be seen as a sequence of matrices indexed by $n \in \mathbb{N}$.

It might be helpful to first define a concrete sequence of multi-distribution learning problems that have natural bit representations. For example, one might define the domain ${\mathcal X}_n = \\{0,1\\}^n$, the collection $(D_1,...,D_k)$ to be distributions whose marginal on $\mathcal{X}_n$ is supported on the standard basis $\\{e_1,...,e_n\\}$, and the baseline class $\mathcal{H}_n$ to be the set of halfspaces on $\mathbb{R}^n$. Then, one can show that for each $n \in \mathbb{N}$, there is an efficient algorithm that maps a discrepancy minimization instance $A \in \\{0,1\\}^{n \times n}$ to a collection of measures (in bit representation) $(D_1,...,D_k)$ which are all efficiently computable and sampleable.

**Motivation for deterministic classifiers?** While I also lean towards deterministic classifiers over randomized classifiers, are there well-founded reasons to prefer deterministic classifiers? It would be helpful if the authors could provide more motivation behind seeking deterministic classifiers.

**Restriction to finite domains for Theorem 2.** In page 3, lines 96-98, the authors say,
> *The restriction to finite domains $\mathcal{X}$ seems to be mostly a technicality in our proofs. Since any realistic implementation of a learning algorithm requires an input representation that can be stored on a computer, we do not view this restriction as severe.*

Could the authors elaborate on why the restriction to finite domains seems to be a mere proof technicality? What are some steps towards establishing the "random rounding" result for continuous domains? What assumptions (e.g., smoothness of the "true" regression function $f^*$) are needed?

In addition, I respectfully disagree with the second part of the quote. For example, sample complexity results in learning theory are independent of whether they can be implemented on a computer or not. Without further investigation, it seems rather premature to claim that the restriction to finite domains is not significant.

**Storage reduction seems unlikely**. I am having trouble understanding how the storage requirement for the derandomized classifier could be reduced at all. Consider the following example. Let $\mathcal{X} = \\{0,1\\}^n$, let the marginal distribution be the uniform distribution over $\mathcal{X}$, and let $f^*(x) = 1/2$ for any $x \in \mathcal{X}$. Suppose the randomized classifier is a uniform mixture of two constant classifiers: all-ones $f_1(x) = 1$ for any $x \in \mathcal{X}$ and all-neg-ones, $f_{-1}(x) = -1$ for any $x \in \mathcal{X}$. Then, the random rounding algorithm outputs a purely random truth table indexed by $\mathcal{X}$. However, random truth tables are incompressible.

**Questions:**

1. What is a formal definition of multi-distribution learning in the uniform computation model? Suppose we want to say a Turing machine M "solves" multi-distribution learning. What are the inputs to M? Is it the collection of distributions $(D_1,...,D_k)$? Does it also include the baseline class $\mathcal{H}$? How are these inputs represented?

2. Are there well-founded reasons to prefer deterministic classifiers over randomized ones?

3. Could the authors elaborate on why the restriction to finite domains seems to be a mere proof technicality? What are some steps towards establishing the "rounding" result for continuous domains?

4. In what sense can the storage requirement be reduced? It seems unlikely that *any* derandomized classifier can be compressed.

---

> ### Author Rebuttal · Authors · 2024-08-05
>
> Thanks a lot for the review.
>
> Let us answer question 1 last as it has the longest reply.
>
> Question 2: See also the discussion with reviewer QSLG  on remark 3. Furthermore, you only have Markov's inequality to bound the probability of performing significantly worse than your expected error. Indeed, the example outline for reviewer QSLG shows an expected error of 1/k, but with probability 1/k over drawing a hypothesis from the distribution, it might be as high as 1. A deterministic predictor on the other hand has a strong performance guarantee on all k distributions simultaneously.
>
> Question 3: Here is one approach that could probably generalize to continuous domains. Observe that if the VC-dimension of the hypothesis set H is d, then the so-called dual VC-dimension d* is at most 2^d. This implies that if the randomized multi-distribution learner that we invoke as a black-box has created a distribution over a set of M hypotheses, then the infinite input domain X really only consists of (M choose 2^d) distinct points (in terms of how they can be labeled by the M hypotheses). It would thus seem that one can restrict to a finite domain of this size when rounding.
>
> Question 4: The important point here is that we are NOT compressing the arbitrary derandomized classifier. Instead, what we show is that there is a small set of derandomized classifiers (those defined from the r-wise independent hash function), such that one of these will have an accuracy that is comparable to the randomized classifier. What this intuitively exploits, is that it is not necessary to independently round each and every single prediction in the input domain X. Instead, we can round them in a dependent way (but still r-wise independent) while maintaining roughly the same accuracy as the randomized classifier.
>
> Question 1:
> We agree with the reviewer that we have not given a completely formal definition of multi-distribution learning in a uniform computation model. This was a choice based on who we thought would be the target audience at NeurIPS. But we'll certainly add such a discussion to the paper (possibly in Appendix). We also note this is related to classical treatments of efficient PAC learning (see e.g., book by Kearns & Vazirani). Here are some further thoughts on the matter:
>
> First, we find it most natural that $\mathcal{H}$ is part of the learning problem, i.e. not an input to the algorithm, but is allowed to be "hard-coded" into the algorithm. This is the best match to standard learning problems, where e.g. the Support Vector Machine learning algorithm knows that we are working with linear models. Similarly, the input domain seems best modeled by letting it be known to the algorithm. One tweak could be that if the input is $d$-dimensional vectors, then $d$ could be part of the input to the algorithm. This again matches how most natural learning algorithms work for arbitrary $d$ (and we need $d$ to grow for our $n$ to grow).
>
> Now regarding modeling multi-distribution learning, we find that the following uniform computational model most accurately matches what the community thinks of as multi-distribution learning (here stated for the input domain being $n$-dimensional vectors and the hypothesis set being linear models).
>
> A solution to multi-distribution learning with linear models, is a special Turing machine M. M receives as input a number $n$ on the input tape. In addition to a standard input/output tape and a tape with random bits, M has a "sample"-tape, a "target distribution"-tape and a special "sample"-state. When M enters the "sample"-state, the bits on the "target distribution" tape is interpreted as an index in $i$ and the contents of the "sample"-tape is replaced by a binary description of a fresh sample from a distribution $D_i$ ($D_i$ is only accessible through the "sample"-state). A natural assumption here would be that $D_i$ is only supported over points with integer coordinates bounded by $n$ in magnitude. This gives a natural binary representation of each sample using $n \log n$ bits, plus one bit for the label.
>
> M runs until terminating in a special halt state, with the promise that regardless of what $n$ distributions $D_1,\dots,D_n$ over the input domain that are used for generating samples in the "sample"-state, it holds with probability at least $2/3$ over the samples and the random bits on the tape, that the output tape contains a binary encoding of a hyperplane with error at most $\tau + 1/n$ for every distribution $D_i$. A bit more generally, we could also let it terminate with an encoding of a Turing machine on its output tape. That Turing machine, upon receiving the encoding of $n$ and an $n$-dimensional point on its input tape, outputs a prediction on its tape. This allows more general hypotheses than just outputting something from $\mathcal{H}$.
>
> Now observe that our reduction from discrepancy minimization still goes through. Given a special Turing machine M for multi-distribution learning, observe that we can obtain a standard (randomized) Turing machine $M'$ for discrepancy minimization from it. Concretely, in discrepancy minimization, the input is the integer $n$ and an $n \times n$ binary matrix $A$. As mentioned in our reduction, we can easily compute $n$ shattered points for linear models, e.g. just the standard basis $e_1,\dots,e_{n}$. Now do as in our reduction and interpret each row of $A$ as two distributions over $e_1,\dots,e_{n}$. M' can now simulate the "sample"-state, "sample"-tape and "target distribution" tape of M, as it can itself use its random tape to generate samples from the distributions. In this way, M' can simulate M without the need for special tapes and states, and by the guarantees of M (as in our reduction), it can distinguish whether $A$ has discrepancy 0 or $\Omega(\sqrt{n})$ by using the final output hypothesis of M and evaluating it on $e_1,\dots,e_n$ and computing the error on each of the (known) distributions $D_i$ obtained from the input matrix $A$.

---

> ### Comment · Reviewer_x5BR · 2024-08-09
>
> Thank you for addressing my questions and being receptive. I've increased my ratings based on the authors' clarifications, especially their responses to my Q1 and Q2.

---

### Official Review · Reviewer_2SLq · 2024-07-11

**Soundness:** 3
**Presentation:** 4
**Contribution:** 3
**Rating:** 7
**Confidence:** 3

**Summary:**

This paper explores the setting of multi-distribution learning in the binary label setting, where the goal is to output a classifier that does well with respect to a set of distributions, rather than just a single distribution. In the agnostic case, all existing learning algorithms output a randomized classifier specified as a distribution over classifiers. The authors investigate to what degree this randomization is necessary, and show that in the most general case, de-randomizing may require super-polynomial training or evaluation time. On the positive side, they show that in the case where the set of distributions shares the same conditional label distribution for each covariate, they can give an algorithm for derandomizing a randomized classifier.

**Strengths:**

- The paper is written very clearly, and the proofs and statements are easy to follow, and seem to be correct.

- The question of hardness of derandomization for multidistribution learning seems well motivated, and the hardness result requires minimal assumptions and is proved through an elegant reduction.

**Weaknesses:**

- It would be great to have some discussion of other related areas where a learning process outputs a randomized predictor that could be derandomized, such as in no-regret learning.

- Similarly, the label-consistent setup seems equivalent to the problem of learning with covariate shift, and comparing the results to prior work in this area would be useful.

**Questions:**

- Is there any reason why the output is restricted to be a classifier rather than a deterministic predictor for the conditional distribution given x? Does allowing predictors to be outputted make the problem significantly easier?

- Is there a reason why you provide sample complexity rather than runtime and evaluation time results for the label-consistent derandomization algorithm? This would be a better comparison to Theorem 1.

**Limitations:**

The authors adequately address the assumptions they make and the limitations of their work.

---

> ### Author Rebuttal · Authors · 2024-08-05
>
> We thank the reviewer for the review.
>
> For question 1 on predictor for the conditional distribution: If we interpret your question correctly, you are asking whether it would be sensible to ask only that on a given x, we output Pr[y = 1 | x] and Pr[y = -1 | x] instead of a prediction of the class. While this is very natural, like in logistic regression, a central question is what happens to the loss function? Should one instead use a logistic loss rather than a 0/1-loss? While this is an interesting direction for future research, multi-distribution learning in this setup would be a completely different problem and a new question in its own right. Whether easier or harder seems difficult to say at this time.
>
> For question 2 on why sample complexity: You are correct that it would have been natural to also mention the runtime. Our focus on sample complexity was due to previous works focussing on sample complexity, and we wanted to demonstrate that one can maintain optimal sample complexity. However, our reduction itself is efficient, except possible for the black-box application of randomized multi-distribution learning. We will thus expand our theorem to say that we can solve the deterministic learning problem with a runtime that equals the randomized version, plus a small (linear in the sample complexity) additive term. Regarding the running time of randomized multi-distribution learning, note that some but not all of the previous algorithms are computationally efficient. The algorithm by Haghtalab et al. uses Hedge on the hypothesis class, and as such is not efficient. However, the more recent algorithm by Zhang et al. is computationally efficient if Empirical Risk Minimisation is efficient for the hypothesis class. This efficiency translates to the deterministic setting by our reduction.

---

> > ### Comment · Reviewer_2SLq · 2024-08-10
> >
> > Thank you for your response to my review! In light of the clarifications you offer and your responses to other reviewers, I will raise my score to a 7.

---

### Official Review · Reviewer_QSLG · 2024-07-11

**Soundness:** 3
**Presentation:** 3
**Contribution:** 3
**Rating:** 6
**Confidence:** 3

**Summary:**

This paper considers the *multi-distribution learning* setting of Haghtalab et al. (2022), where, given $k$ unknown data distributions $\\{\\mathcal{D}_1, \\dots, \\mathcal{D}_k\\}$, the goal is to find a classifier $f: \\mathcal{X} \\rightarrow \\{-1, 1\\}$ such that $f$ performs almost as well as $\underset{h \in \mathcal{H}}{\min} \underset{i \in [k]}{\max} \underset{{(x, y) \sim \mathcal{D}_i}}{\mathrm{Pr}}[h(x) \not = y]$ on every distribution $\mathcal{D}_i$ for $i \in [k]$.

The paper has two main results. The first is a negative result that states that, under mild conditions on $\\mathcal{H}$, any learning algorithm on $\\mathcal{H}$ that produces a classifier $f$ satisfying multi-distribution learning must have either super-polynomial training time or $f$ must have super-polynomial evaluation time. This is done through a reduction to the NP-Hard problem of Discrepancy Minimization. The second is a positive result that states that, under the condition of *label-consistency* (the conditional distributions of all the distributions agree, for all $x \\in \\mathcal{X}$) and finite input space, there does exist a learning algorithm that produces a deterministic classifier $f: \\mathcal{X} \\rightarrow \\{-1, 1\\}$ satisfying multi-distribution learning. It produces this classifier via black-box access to an existing multi-distribution learning algorithm that outputs a randomized classifier.

**Strengths:**

Overall, this paper is well-written and clear. I had no trouble following along with the arguments, and each proof is sketched out nicely before getting into the details. I believe the result does address an important issue in the multi-distribution learning literature, as I have personally found the fact that every existing multi-distribution learning relies on a randomized classifier predicting with a rather opaque distribution (usually obtained by running a no-regret algorithm) unsatisfying.

**Originality:** The paper is original insofar as it provides the first (to my knowledge) derandomized classifier for the agnostic multi-distribution learning problem. I do wonder whether the algorithm itself can be made simpler, as it seems somewhat artificial to me to explicitly designate the outputs of our final classifier (defined as $\hat{f}$) one-by-one through enumerating part of the input space $\mathcal{X} \setminus T$. That being said, taking the result as a purely a proof of existence, the work is novel and addresses an important question.

**Quality:** The quality of the paper itself is good -- there are few typographical issues, the structure of the paper is clear, and the theorems are clearly presented.

**Clarity:** There is no issue with clarity -- the authors have formatted the paper well, and the arguments are clearly presented.

**Significance:** I believe the de-randomization question of multi-distribution learning to be significant, though more applied practitioners may view the paper as too theoretical. On the grounds that this is a learning theory work, however, this addresses an important question presented in COLT 2023 by Awasthi et al. (Open Problem: The Sample Complexity of Multi-Distribution Learning for VC Classes) through demonstrating a statistical-computational gap.

**Weaknesses:**

One potential weakness in the work is that the final deterministic classifier relies on explicitly defining the outputs for all $x \not \in T$ by sampling from the existing multi-distribution learner's distribution. I don't find this to be too much of a weakness, as I do think that it is interesting that a deterministic algorithm *exists* in the first place, but I imagine that this might be viewed as a strong assumption. Perhaps the authors can provide some text motivating this deterministic classifier and its somewhat unwieldy final form (maybe this can be done by a brief discussion on the size of $T$?).

Another point that the authors could address a bit more is how natural the assumption of label-consistency for the positive result is. There is a brief reference to Ben-David and Urner (2014), but it would be helpful to include a further discussion on this assumption in the context of PAC learning. It seems that this is closely connected to the predominant assumption in the *covariate shift* literature, but a further discussion on perhaps the results of Ben-David and Urner (2014) in the body of this paper would be helpful to judge the strength of this main assumption. This is especially important because we are considering it in the context of multi-distribution learning.

Finally, there are a couple of suggestions I had that may clarify the presentation of the paper:

1. It might be a bit clearer to define $\underset{\mathrm{er}}{\mathcal{P}}$ on a separate line (not in Equation 1). Personally, I took a while trying to find where $\underset{\mathrm{er}}{\mathcal{P}}$ was defined in the paper, and it's an important quantity.
2. In the Introduction, I would've appreciated some quick details on the existing algorithm for the deterministic classifier in the realizable setting. A one or two sentence sketch would be sufficient.
3. It is unclear to me why the claim just above "Our contributions" about using the random classifier and Markov's inequality is the "best we can guarantee." Can you justify that this is indeed "the best" argument for using a classifier drawn from the distribution $F$?
4. I was initially confused about which definition you were taking for *label-consistent learning*, as you write, "As a particular model of label-consistent learning, one may think of the deterministic labeling setup..." so initially I thought you were restricting the assumption for the positive result to the deterministic labeling case with $f^*: \mathcal{X} \rightarrow \{-1, 1\}$. I would make clear (perhaps in a Definition environment) the label-consistent learning definition is, rather, the $\mathcal{D}_i(y \mid x) = \mathcal{D}_j(y \mid x)$ for all $i, j, x$ assumption.
5. Small nitpick: In Section 2, you don't define the matrix $A \in \mathbb{R}^{n \times n}$ before presenting the definition of Discrepancy Minimization.

**Questions:**

I had two major questions about the work:
1. I was wondering about was whether it is possible to do such a de-randomization from scratch, i.e. without assuming black-box access to an existing multi-distribution learning algorithm. I wonder this because existing algorithms for multi-distribution, to my knowledge, all rely on applying Hedge (or a similar no-regret algorithm) to the $k$ distributions, which can be potentially expensive in situations with many distributions. I wonder if there are techniques to directly obtain a de-randomized classifier, potentially following the lines of the deterministic classifier for the realizable case.
2. Did the authors consider any other assumptions besides label-consistency that might allow for a deterministic algorithm?

These last two questions aren't criticism on the work itself -- I was just curious if the authors thought about these issues.

One other thing that might be worth mentioning is that [Zhang et al. (2023)](https://arxiv.org/abs/2312.05134), Theorem 2 also presents a hardness result on derandomizing multi-distribution learning. However, their result is a statistical hardness result, while yours is computational. It would be helpful to put this paper in context perhaps with a couple of sentences in the Introduction.

**Limitations:**

The authors have addressed the limitations of the work on a theoretical basis in the main text (see "Discussion of Implications"). Because this is mainly a theory paper, negative social impacts or broader impacts to applications seem minimal.

---

> ### Author Rebuttal · Authors · 2024-08-05
>
> Thanks a lot for your review.
>
> To answer your questions:
>
> For remark 3 regarding "best we can guarantee": Well, from the guarantees of a randomized multi-distribution learner alone, the claim in 42-44 indeed seem to be roughly the best possible. Consider for instance k distributions on a domain X of cardinality k. The labels of all k points in the domain are deterministically 1. There are k data distributions, where $D_i$ has all of its probability mass on the $i$'th point of the input domain. There are k hypotheses, each returning 1 on all but a single point where it returns -1. If we look at the randomized predictor that is uniform random over these k hypotheses, then for each $D_i$, the expected error is $1/k$. However, if we draw a hypothesis from the randomized predictor, then it will have error 0 on k-1 of the distributions and error 1 on one of them. Thus a factor k worse, i.e. Markov is the best we can hope for.
>
> For Question 1 regarding de-randomization from scratch: Indeed this would be possible to attempt. Note however that by giving a general and efficient black-box reduction from randomized multi-distribution learning, we have made it easier to obtain good deterministic classifiers. Indeed, since it is easier to obtain a randomized one, we can just start with that. However, aiming directly for a deterministic classifier to begin with seems like an interesting direction for future research.
>
> For Question 2 regarding other definitions: As also mention in our reply to reviewer xqfu and in lines 75-79, we came up with our definition of label-consistency by examining the NP hardness result. Concretely, we found the most general definition (least restrictive) that allowed us to circumvent the lower bound. As such, we did not consider other notions. As can be seen in our lower bound, the critical property of the lower bound instance is that the same input point can be assigned label -1 in one distribution and +1 in the other (both with probability 1). Any useful definition must at least prevent this, and our label-consistency seemed the most natural and least restrictive such definition.
>
> > One other thing that might be worth mentioning is that Zhang et al. (2023), Theorem 2 also presents a hardness result on derandomizing multi-distribution learning. However, their result is a statistical hardness result, while yours is computational. It would be helpful to put this paper in context perhaps with a couple of sentences in the Introduction.
>
> We discuss this at the end of Introduction in lines 99-110.

---

> > ### Comment · Reviewer_QSLG · 2024-08-09
> >
> > I appreciate the authors' clarifications, and I have read the response. Thank you for clarifying my question about the "Markov is the best we can hope for" -- this makes sense now. I'd like to keep my score and overall positive evaluation of the paper.

---

### Official Review · Reviewer_xqfu · 2024-07-17

**Soundness:** 4
**Presentation:** 3
**Contribution:** 3
**Rating:** 6
**Confidence:** 2

**Summary:**

The paper studies the problem of multi-distribution learning in the agnostic setting. The authors provide a a computational lower bound for deterministic classifiers (based on some mild assumption on the hypothesis class, and the assumption $\text{BPP}\neq \text{NP}$), as well as an upper bound: an improper deterministic classifier for a special case of label-consistent distributions (i.e. when $D_i(y | x) = D_j(y | x)$ for all $i,j,x$). Their classifier uses a randomized learner as a black box, and its sample complexity is just almost the same as the sample complexity of the randomized learner.

**Strengths:**

The results look interesting to me, they are non-trivial and novel (though I'm not an expert in the field, so I might have a wrong impression). The assumption of label-consistency looks natural to me, though it is not clear to me how restrictive it is.

**Weaknesses:**

While the results look new and non-trivial, I am not sure how strong the technical contribution is. I did not check the proofs in detail, and at the first glance the ideas don't look very sophisticated.

**Questions:**

As I already mentioned, it is not clear how restrictive label-consistency is. I'm not an expert in the area, and this assumption looks natural to me, but how is it related to standard assumptions? Was it already studied in prior works?

**Limitations:**

The authors adequately addressed the limitations.

---

> ### Author Rebuttal · Authors · 2024-08-05
>
> Thanks a lot for your review. Regarding label consistency, please note that lines 81-86 compare it to the previous definition of deterministic labels by Ben-David and Urner. According to their results, in the case of a single distribution, deterministic labels are statistically almost as hard as the general agnostic case. Clearly our definition is more general and thus our results at least as strong. Furthermore, notice that our definition of label-consistency reduces to the precisely the standard agnostic PAC learning setup when there is just one data distribution. Let us also add that our definition was inspired by the lower bound construction, see lines 75-76. We made the most general definition of data distributions that we could come up with that still allowed circumventing the hardness results. To the best of our knowledge, the exact definition we use has not been studied before. However, this might not be too surprising as it only makes sense in a multi-distribution setting.

---

### Decision · Program_Chairs · 2024-09-25

**Decision:**

Accept (poster)

**Comment:**

This paper studies the recently introduced talk of multi-distribution learning, where roughly speaking the goal is to find a classifier that performs well simultaneously over multiple distributions. Prior work had shown how to obtain randomized classifiers for this task with sample complexity proportional to the VC dimension of the base class. The current work gives two results: (1) First it is hard to construct a deterministic classifier in the general case, and (2) if an additional condition, termed label consistency, is satisfied one can obtain efficiently a deterministic classifier. The consensus from the reviews and discussion was that this is a technically worthy contribution that should be accepted to the conference.